# Geographic Patterns of the Richness and Density of Wild Orchids in Nature Reserves of Jiangxi, China

**Qinghua Zhan** [1,2,†], **Yuelong Liang** [3,†], **Zhong Zhang** [4,†], **Feihu Liu** [1,2], **Liyang Li** [1,2], **Xiaodong Tang** [1,2], **Zhongxuan Liang** [1,2], **Weixing Chen** [1,2], **Mingtao Hu** [1,2], **Shaolin Tan** [1,2], **Huolin Luo** [1,2], **Yadong Zhou** [1,2] and **Boyun Yang** [1,2,*]

1   School of Life Sciences, Nanchang University, Nanchang 330000, China
2   Key Laboratory of Plant Resources in Jiangxi Province, Nanchang 330000, China
3   Jiulianshan National Nature Reserve of Jiangxi, Ganzhou 341700, China
4   Jinggangshan National Nature Reserve of Jiangxi, Ji'an 343000, China
*   Correspondence: yangboyun@163.com
†   These authors contributed equally to this work.

**Abstract:** Orchids have attracted much attention from researchers, because of their richness of species and their great ornamental and medicinal value. Jiangxi Province, which is located in southeastern China and ringed on three sides by mountains, contains many nature reserves and harbors large number of orchids. Here, we conducted field surveys of orchids in 35 nature reserves in Jiangxi, using sampling lines and plots. We also analyzed the relationship between orchid richness and density with environmental variables and studied the relationship among these nature reserves. We found that the mountainous areas of southwestern, southern, and northeastern Jiangxi have a high richness and density of orchids, while the mountainous areas of central and northwestern Jiangxi have low richness and density. Jiulianshan and Jinggangshan are the two most rich-species reserves, with 58 and 55 orchids, respectively. Eight reserves (22% of those surveyed) had fewer than 10 orchids. Compared with soil, climate, and vegetation, topography was more closely related to the richness and density of orchids. Topographical variables explained 19% and 20% of the total variation in SR and SD, respectively. The result of hierarchical clustering analysis showed that the 35 nature reserves of Jiangxi obviously fall into two main clusters, which are separated by the Ganjiang River–Poyang Lake water system. In conclusion, the geographical patterns of richness and the density of orchids in Jiangxi are uneven and are affected by topography and vegetation, while their distribution is affected by the terrain of Jiangxi. Our work explains the richness and density patterns and the assembly mechanism of the orchids in Jiangxi and also provides new ideas for the protection of orchids in this region.

**Keywords:** Orchidaceae; investigation; diversity; mountains; Poyang Lake basin

## 1. Introduction

The family of orchids (Orchidaceae) is one of the largest families of flowering plants in the world, with approximately 750 genera and 28,000 species, and contains a large number of critically endangered species [1]. Orchids are widely distributed in various terrestrial ecosystems around the world, especially in tropical regions, except for polar and arid desert regions [1–3]. China is one of the countries with the richest diversity of orchids in the world [4]. Currently, there are approximately 1708 orchids recorded in China, belonging to 181 genera, 17 tribes, and 5 subfamilies [4].

At present, the understanding of biodiversity is still insufficient. Many new species are discovered and published every year [5]. Orchidaceae is one of the families with the richest discovery of new taxa by researchers [5–7]. For example, a total of 20 species, 1 hybrid, 6 varieties, and 1 form of Orchidaceae have been discovered in China in 2020 [6]. The main reasons may be that orchids are highly diverse, with small plants and a short flowering

period, and most of them grow in an undisturbed environment, such as epiphytic orchids that grow on trees or rock walls. In the early stages, researchers had difficulties in obtaining complete species inventories.

Environmental variables affect the propagation and growth of plants and could determine the distribution of plant species to a great extent. Previous studies showed that the distribution of orchids was regulated by habitat size and elevation range at large scales, while it was regulated by light availability, soil moisture, and vegetation at small scales [8–11]. Research based on the literature, floras, specimens, and other empirical data showed that net primary productivity, elevation range, and temperature seasonality together explained 34.4% of variance in orchid richness in China [11]. However, research based on field survey data showed that vegetation type, elevation, aspect, slope, mean annual temperature, and precipitation could only explain 3.7% of the variation in the composition of wild orchid plants on Hainan Island, China [12].

Jiangxi is located in southeastern China, with its geographical range roughly overlapping the Poyang Lake basin [13]. There are more 190 nature reserves at different levels in Jiangxi, including 16 national and 39 provincial nature reserves. These nature reserves harbor a large number of wild orchids, but the field investigation for orchids in these reserves is incomplete [14]. For example, there are only 69 orchids of Jiangxi recorded in *Flora of China* [15]. However, the latest statistics showed that there are more than 100 orchids in Jiangxi [14], and this number has increased to 193 after several years of investigation by our team [16–19]. There have been a lot of new records and new taxa reported in recent years, such as *Danxiaorchis yangii* B.Y.Yang & Bo Li [20] and *Calanthe sieboldopsis* B.Y.Yang & Bo Li [21] from Jinggangshan National Nature Reserve. Here, based on the survey data from thousands of plots in 35 nature reserves of Jiangxi, we studied the orchid diversity in this region. Specifically, we intend to address the following questions: (1) What are the geographic patterns of the richness and density of orchids in the nature reserves of Jiangxi? (2) Which environment variables are most important in affecting the orchids in Jiangxi? (3) How are the mountain ranges of Jiangxi affecting the distribution of orchids in this region?

## 2. Materials and Methods

### 2.1. Study Area

Jiangxi is located in the southeastern China, bordered by Zhejiang and Fujian in the east, Guangdong in the south, Hunan in the west, and Hubei and Anhui in the north. Jiangxi is surrounded by mountains in the east, west, and south, and Poyang Lake plain in the north [13]. The main mountain ranges in the territory include Wuyi Mountains and Huaiyu Mountains in the east; Luoxiao Mountains, Mufu Mountains, and Jiuling Mountains in the west; and Dayu Mountains and Jiulian Mountains in the south (Figure 1). Ganjiang River flows through Jiangxi from south to north and flows into Poyang Lake. This water system divides the main mountain ranges of Jiangxi into east and west parts and may have geographical impact on the distribution of plants.

### 2.2. Field Surveys

Our field investigation in 35 nature reserves of Jiangxi lasted for two years, from 2021 to 2022. These reserves almost cover all the main mountains of Jiangxi (Figure 1). The places with rich orchids were chosen for setting up the sampling lines with a length greater than 1000 m; then, 1 to 20 5 m × 5 m plots were set up on both sides of the sampling lines. The criteria of setting plot are that plots should harbor orchids and distance between two plots is greater than 10 m. In total, more than 2400 plots were set up in the 35 nature reserves (Figure 1; Supplementary Material Table S1). Geographical coordinates and orchid species were recorded for each plot. In order to protect orchids, we did not collect specimens from every orchid that was encountered. We just took pictures for each orchid and collected samples and specimens for some species that cannot be identified in the wild. Samples and specimens were used to accurately identify those unknown species in the wild.

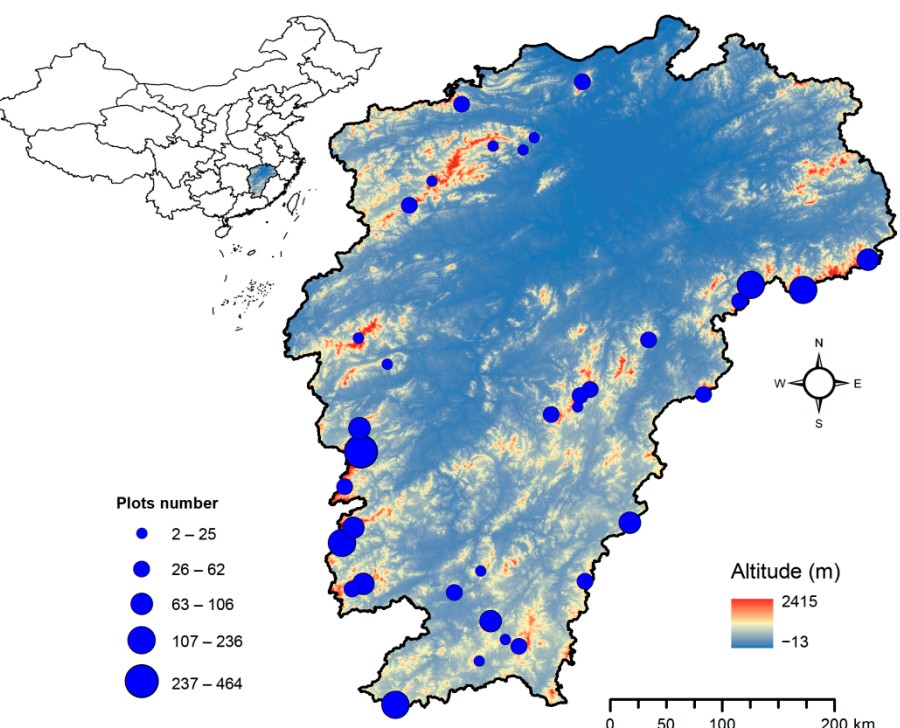

**Figure 1.** The locality of the 35 nature reserves of Jiangxi.

*2.3. Environmental Variables*

Ten environmental variables of these plots were used for analysis in this study, including topography (elevation and slope), vegetation (canopy density (CD) and normalized difference vegetation index (NDVI)), soil (pH, total nitrogen (TN), total phosphorus (TP), and total potassium (TK)), and climate (annual mean temperature (AMT) and annual precipitation (AP)). Elevation, slope, and CD were measured in the field, and other variables were extracted from raw data sets using ArcGIS 10.5 [22]. The raw data sets of NDVI during 2011 to 2020 were downloaded from Resource and Environment Science and Data Center (https://www.resdc.cn/, accessed on 20 August 2022), soil variables were downloaded from National Tibetan Plateau/Third Pole Environment Data Center (http://www.tpdc.ac.cn/, accessed on 20 August 2022) [23], and climatic variables were downloaded from WorldClim (http://www.worldclim.org/, accessed on 5 May 2021). The environmental variables of each nature reserve were calculated by the mean values of the variables among all the plots located in the nature reserve (Table S1).

*2.4. Data Analysis*

Based on the data matrix of orchids' present or absent in the 35 nature reserves of Jiangxi, we calculated the species richness (SR) of orchids in these reserves. There is always a positive correlation between SR and the area of nature reserves, so we also calculated the species density (SD) of orchids for each nature reserve using the formula $SD = SR/\ln(A)$ [24,25], where A is the area of each nature reserve. Linear regressions were used to explore the relationship between SR or SD and all the environmental variables. Variance partitioning analysis was used to examine the contribution of total, shared, and independent effects of the four groups of environmental variables on SR and SD of orchids. Before performing variance partitioning analysis, the correlations of pairwise variables were tested using Pearson correlation, and all these variables did not exhibit strong correlation ($|r| < 0.70$) [26]. In order to find out the relationships among nature reserves, we used Jaccard's index to calculate the taxonomic β-diversity of orchids and used Ward's method to carry out the hierarchical clustering analysis [27]. All the analyses were carried out using the R software (Version 3.3.3) [28].

## 3. Results

### 3.1. Richness and Density of Orchids in Nature Reserves of Jiangxi

In total, 125 orchids belonging to 55 genera were found by us in the 35 nature reserves of Jiangxi, including 85 terrestrial (including 10 mycoheterotrophic) and 40 epiphytic species (Table S2). Of these orchids, 50 species belonged to the least concern (LC) category of the IUCN Red List, 13 to the near threatened (NT) category, 17 to the vulnerable (VU) category, 11 to the endangered (EN) category, and 5 to the critically endangered (CR) category (Appendix S2). *Bulbophyllum* Thou. and *Calanthe* R. Br., both with 10 species, are the two most species-rich genera, while over half of these genera only have one species (Figure 2). Based on our field work, there were 23 new records of orchids in Jiangxi (Table S2). Reserves with high SR and SD of orchids are mainly distributed in the reserves of the mountain areas around Jiangxi, while these are low in the reserves of the mountain areas in central and northwestern Jiangxi (Figure 3). Jiulianshan and Jinggangshan are the two most species-rich reserves, with 58 and 55 orchids, respectively. Eight reserves (22% of those surveyed) had fewer than 10 orchids (Figure 3; Table S2).

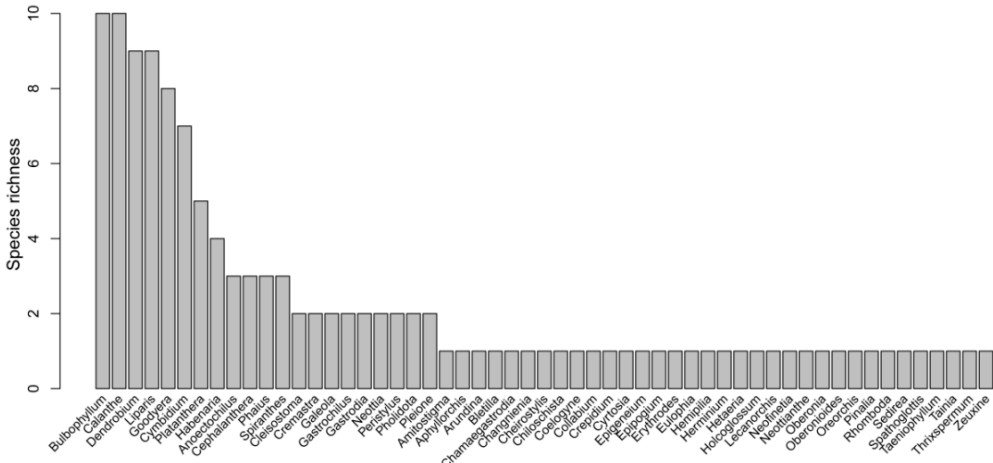

**Figure 2.** Species richness of each genus of orchids.

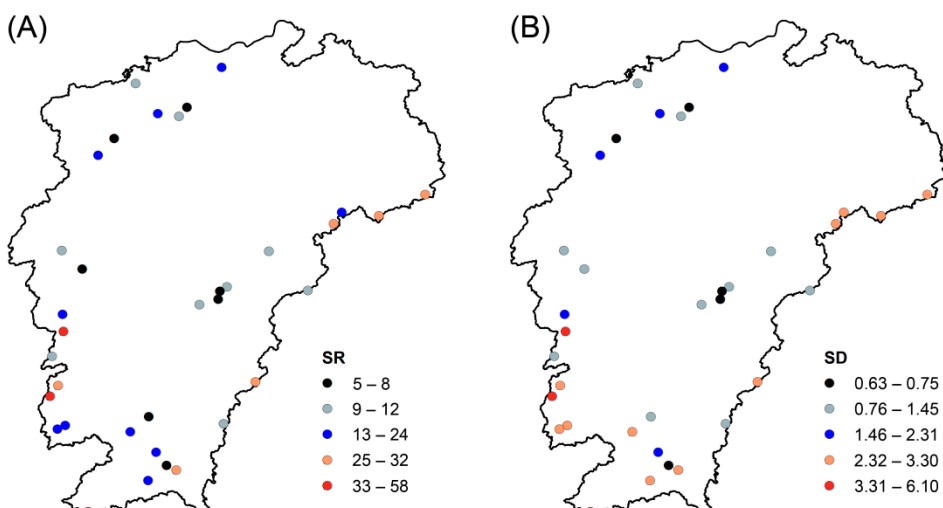

**Figure 3.** Geographic patterns of (**A**) species richness (SR) and (**B**) species density (SD) of orchids in 35 nature reserves of Jiangxi.

### 3.2. Effects of Environmental Variables on Richness and Density of Orchids

Except for slope having significant positive correlation with SR and SD (Figure 4A,B), and NDVI having significant negative correlation with SR (Figure 4C), other environmental variables have no relationship with both indexes. The results of variance partitioning

analysis showed that these four groups of environmental variables could not explain the richness and density of orchids very well, i.e., they only together explained 22% or 26% of the variation in SR and SD of orchids (Figure 5). Topographical variables explained more variation of both SR and SD than the other three groups of variables. Topographical variables explained 19% and 20% of the total variation in SR and SD, respectively (Figure 5).

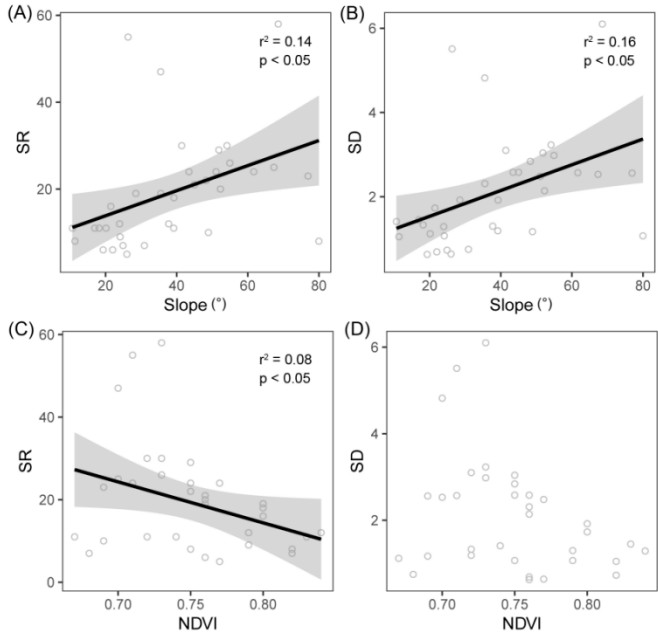

**Figure 4.** Linear relationships between species richness (SR) and species density (SD) with slope and normalized difference vegetation index (NDVI). (**A**) relationship between SR and slope; (**B**) relationship between SD and slope; (**C**) relationship between SR and NDVI; (**D**) relationship between SD and NDVI.

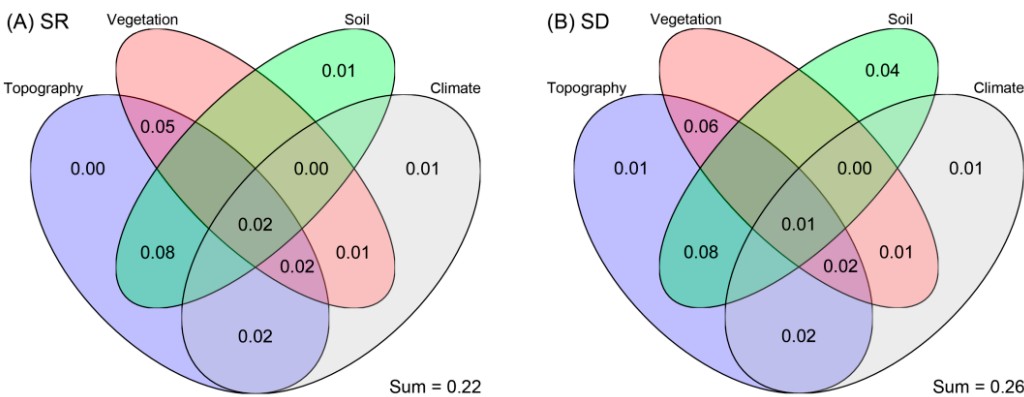

**Figure 5.** Variation partitioning of four environmental variable groups for (**A**) species richness (SR) and (**B**) species density (SD) of orchids in Jiangxi. The four groups of environmental variables are topography (elevation and slope), vegetation (canopy density and normalized difference vegetation index), soil (pH, total nitrogen, total phosphorus, and total potassium), and climate (annual mean temperature and annual precipitation).

### 3.3. Hierarchical Clustering of Nature Reserves

The result of hierarchical clustering analysis showed all these 35 nature reserves fall into two main clusters (A and B) (Figure 6). Cluster A contained most of the nature reserves from eastern and southern Jiangxi, while cluster B contained the nature reserves from western and northwestern Jiangxi; Ganjiang River and Poyang Lake could separate the two clusters geographically (Figure 6).

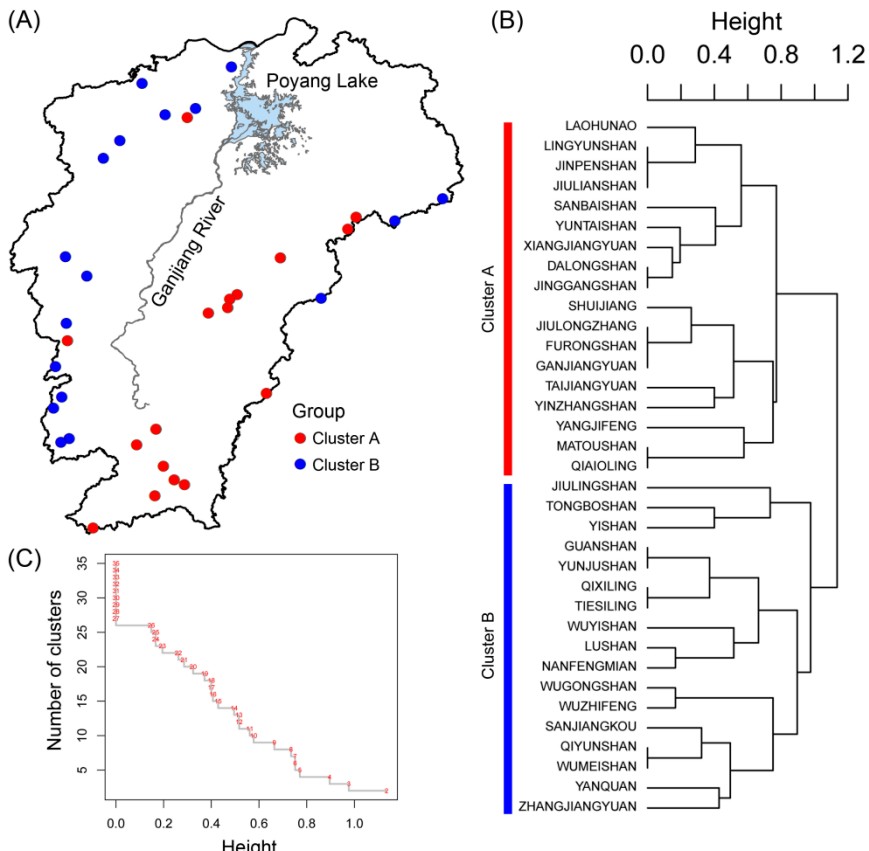

**Figure 6.** Map (**A**), dendrogram (**B**), and fusion level (**C**) resulting from ward hierarchical clustering of 35 nature reserves of Jiangxi, based on the similarity of the reserves in terms of orchid richness.

## 4. Discussion

Jiangxi is surrounded by mountains on three sides and basically coincides with the Poyang Lake basin [14]; Jiangxi has a complex terrain, developed water systems, and a warm and humid climate, which breeds rich biodiversity [29]. However, the field survey of plant diversity in Jiangxi is incomplete, especially for some species-rich large families, such as Orchidaceae. Our field surveys showed that there are 125 species of orchids in only 35 nature reserves, including 23 new records of Jiangxi. This work fully showed that the investigation of orchids in Jiangxi is not comprehensive, especially for some mountains with high species richness, such as the Wuyi Mountains and the mountains of southern Jiangxi [14].

Currently, there are more than 1700 orchid species recorded in China, and *Bulbophyllum* Thou. and *Dendrobium* Sw. are the two most species-rich genera, each with more than 100 species [4]. Both of the two genera are also rich in species in Jiangxi, based on our investigation. The areas rich in orchids are concentrated in southwest, south, and northeast Jiangxi, though are relatively low in central and northwest Jiangxi. This is basically consistent with the results of previous studies [18]. For example, a previous study based on a summary of literatures and field surveys showed that there are more than 120 species of orchids in the Luoxiao Mountains in southwest Jiangxi [30].

A positive species-area relationship is one of the most robust summaries in ecology [31]. We also found that larger nature reserves harbor more orchid species. After removing the area effect, we found that the species density and species richness of orchids have similar geographical patterns in Jiangxi. However, some reserves in south Jiangxi have few species, but the species density is relatively high in, for instance, Zhangjiangyuan, Taojiangyuan, and Yuntaishan. Our results showed that environmental variables do not explain orchid richness and density very well, and this result was consistent with that of

Hainan Island [12] and also the whole country of China [11]. However, we found that topography, especially slope, was closely related to orchid richness and density, rather than other environmental variables, such as soil and climate. This was different from the results of previous studies [12] and indicated that the complexity of terrain can affect orchid richness and density more than climate, on small geographical scales [32].

Most previous studies of the orchids in Jiangxi considered the impact of mountains on the diversity pattern, but ignored the role of water systems [13,16,18]. The mountains of Jiangxi generally stretch from north to south, while the Ganjiang River and Poyang Lake divide the main mountain ranges of Jiangxi into east and west groups. The distances between the mountain ranges of Jiangxi from east to west are far greater than those between the mountain ranges from north to south, and this terrain could cause certain obstacles to the interaction between mountain plants on both sides. We conducted cluster analysis on 35 nature reserves based on orchid distribution information. Our results strongly supported this speculation, i.e., the 35 nature reserves could obviously fall into two main clusters, and the boundary between these two clusters roughly coincided with the Ganjiang River–Poyang Lake water system. Southern Jiangxi is close to the Nanling Mountains, which are considered as an important glacial refuge for plants [33,34]. We speculated that orchids took refuge in the warmer regions of the Nanling Mountains during the glacial period, expanding northward along two different mountain groups after the end of the glacial period [18]. This inference has been confirmed by the studies of population genetic differentiation of some species, such as *Loropetalum chinense* (R. Br.) Oliver [33], *Cercis chuniana* Metc. [34], and *Machilus pauhoi* Kanehira [35].

Orchids are of great interest because of their high ornamental and medicinal value, and their survival is also seriously threatened by climate change, habitat degradation and fragmentation, human disturbance and excavation, etc. [4,11,36,37]. In our study, we did not include human disturbance variables for analysis, but we found that larger reserves harbor more orchids, including some rare and endangered mycoheterotrophic orchids, such as *Danxiaorchis yangii* in Jinggangshan National Nature Reserve [20]. Our survey showed that more than 40 orchids appear in only one of the 35 nature reserves. For example, the vulnerable species *Goodyera bomiensis* K.Y. Lang was only found in Sanbai Mountain National Forest Park. Thus, increasing connectivity between small protected areas and habitats may be beneficial to the protection of orchid diversity [38].

## 5. Conclusions

In this study, we conducted field surveys of orchids in 35 nature reserves in Jiangxi Province, China. We found that the mountainous areas of southwestern, southern, and northeastern Jiangxi have a high richness and density of orchids, while few orchids were found in the central and northwestern Jiangxi. We also found that topography is closely related to the richness and density of orchids, rather than soil, climate, and vegetation. The 35 nature reserves, representing most of the mountain ranges of Jiangxi, were obviously clustered in two main groups, which were also separated by the Ganjiang River–Poyang Lake water system. This indicates that terrain plays an important role in the dispersal of orchids in Jiangxi. Meanwhile, we also put forward a suggestion for the protection of orchid diversity in these regions, i.e., some small protected areas could be integrated to increase the ecological connectivity.

**Supplementary Materials:** The following supporting information can be downloaded at: https://www.mdpi.com/article/10.3390/d14100855/s1. Table S1: Environmental variables of the 35 nature reserves in Jiangxi Province, China; Table S2: Checklist of orchids in the 35 nature reserves of Jiangxi Province, China.

**Author Contributions:** Data curation, Q.Z., M.H., Y.Z. and B.Y.; formal analysis, M.H. and Y.Z.; funding acquisition, B.Y. and Y.Z.; investigation, Y.L., Z.Z., F.L., L.L., X.T., Z.L., W.C., M.H., S.T., H.L. and B.Y.; methodology, M.H. and Y.Z.; supervision, B.Y.; writing—original draft, Q.Z. and Y.Z.; writing—review and editing, Q.Z. All authors have read and agreed to the published version of the manuscript.

**Funding:** This work was supported by grants from the National Forestry and Grassland Administration of China (No. 2021070704) and the National Science Foundation of China (No. 32260046).

**Institutional Review Board Statement:** Not applicable.

**Informed Consent Statement:** Not applicable.

**Data Availability Statement:** Data are available in supplementary files.

**Acknowledgments:** We would like to thank Antony Njogu Waigwa from East China Normal University for revising the English of this manuscript. Many thanks should also be given to the students from Nanchang University and the staff from the nature reserves of Jiangxi for their kindly help with the field surveys.

**Conflicts of Interest:** The authors declare no conflict of interest.

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
