# Peer review of "Geographic Patterns of the Richness and Density of Wild Orchids in Nature Reserves of Jiangxi, China"

_diversity, doi:10.3390/d14100855_

Round 1
Reviewer 1 Report
1.Lines 13,53, 55:Jiangxi located in central China or Southeast China??
2.Lines 15 & 246: Large scale?How Large?
3.Line 54: superior natural environment?
4.Line 38: "found" new species?"discover and pubilish new species" should be better.
5.Line 238:"Danxiaorchis yangii" should be Italic.
Reviewer 2 Report
This paper presents the results of a very labour-intensive, long-term field work aiming to assess the species richness of orchids in province Jiangxi, China. Orchids are a highly diverse and species-rich group of plants, yet highly sensitive to environmental changes and over-collection. While new species are still being described, many are becoming extinct at an alarming rate. Therefore, efforts leading to an exploration of the orchid flora, especially in less surveyed areas, is a valuable initiative.
However, the work has some flaws that need to be corrected before publication. (1) One of them, which requires attention, is the quality of the English language. There are so many grammatical errors that I have resigned myself from correcting them. (2) The authors use the terms "diversity and "richness" interchangeably, which is a mistake. Diversity combines richness and abundance, and the authors use methods that allow only 'richness' to be studied and assessed This should be corrected throughout the work. (3) Richness rarefaction analyses were not performed in this study. This one is necessary to avoid bias caused by unevenly distributed sampling effort. Performing a comparison of richness between nature reserves requires a standardisation of the sample to account for sampling effort. The same applies to richness metric used to perform corelation between environmental variables (4) The authors do not state what are life forms of studied orchid species (i.e., epiphyte, terrestrials). Approximately half of each form is represented in the study, so analyses should take this fact into account, as it may account for poor or no correlation of richness and environmental variables (e.g. soil parameters will be a poor predictor of epiphyte species richness). (5) Furthermore, it is difficult to assess the validity of the statistical analyses, as the authors use parametric tests (Pearson’s correlation) without reporting whether the data have a normal distribution (or whether they have been transformed to meet the assumptions of the test). Moreover, test results are reported without information on whether they were significant or not.
Below are presented my other comments to the manuscript.
The title:
Because the authors recorded only species per plot (richness) and not species abundance , diversity was not assessed (Diversity is richness and abundance). Therefore the title should be changed accordingly.
Abstract is too general and uninformative. Authors should be more specific, key results (specific data) should be placed here to support the general statements.
L12: abundant species change for richness of species
L12-13 Adjust font size.
L13-14 Change for “Jiangxi province is located”.. incorporating another geographical name is unnecessary here. Focus on explaining more explicitly why the authors have selected this region to study orchids.
L 17 What nature of relationship the authors were looking for?
L17-19. The authors should be more consistent in the way they describe regions. If you add the description "mountainous" for one part of the region then what about the other? Is it a lowland area? This is more informative than offering just geographical location - at least in the summary. Here you should also add hard data – richness value etc.
L 19-20 Instead of …? What do you mean by this? Correct the sentence.
L 20 You mean The reserves? The whole sentence is written ungrammatically. Be more specific - what parameter was responsible for grouping the reserves?
Introduction:
L30 It sounds like the polar deserts are in the tropics - correct the syntax.
L53-63 This description of geographic placement is too detailed. General landform would be sufficient. The detailed information of the area of study, factors that may influence species distribution and richness ( geography, topography, climate etc) along with up to date information about orchid flora of the region should be given in Materials and Methods in the subchapter “Study area”
L72-74 the aims definitely needs refinements.
M&M
L 78 – instead “greater than ..” and “several to many” give min-max range. Were plots spaced regularly on lines? If not what was the criterion of their placement? I would suggest adding the time period (year(s), months, seasons, etc.) when the samples were taken and whether the plot lines were visited once or several times to take into account the different flowering periods of the species.
L 84-85 move the sentence right after L.79 - …sampling lines. I would also give here the number of kilometres.
L91-102 The information here is provided in an unstructured manner. The authors should group them appropriately into: field, ArcGIS, other databases etc and assign them explicitly.
Results
Please support the text with quantitative data. These should be given consistently alongside and those presented with figures.
L121-122 Change for: 125 orchid species belonging to 55 genera were found in …..
L126 Authors should try to describe the geographical structure of orchid richness in relation to the landforms of the area, rather than in relation to administrative boundaries, such as 'around Jiangxi'.
L 139 SR and SD are not diversity indexes.
Discussion
L184- 192 The literature information given in this paragraph should be presented in the introduction as a justification for choosing this province to study orchid richness and to present the current state of knowledge.
L219 I do not think that the river system in itself is a real obstacle to orchid seed migration. It could simply be a lack of suitable habitat over long distances between mountain ranges. The sentence just needs rephrasing.
L238 should be mycoheterotrophic instead of saprophytic.
Figures:
Fig 1. The map is not clear. The multiple grey quadrats crowded are not good representation of works done. I’d suggest to place circles with numbers of plots per a given area or if numbers will be too small than use a colour gradient per a given scale of numbers with a legend.
Fig. 3 Please refer to A and B maps in the caption. Yellow dots are poorly visible.
Fig.5 This calculation of the Jaccard coefficient is based on the similarity of the reserves in terms of orchid richness and not on the basis of 125 orchids. Please get the figure description correct.
Appendix S2: Life form of listed orchid species should be given.
Reviewer 3 Report
Comments to the Author
Review of “Diversity and geographic pattern of wild orchids in nature reserves of Jiangxi, China”, by Zhan et al.
This paper has organized very well. It gives us the orchids distributional and diversity pattern in Jiangxi province, and show us the relationship between orchid diversity and environmental variables. This genus involves a large number of endangered species, I suggest the authors show the endangered category and conservation status of orchids in Jiangxi province. And then map them to check if there is some pattern of the endangered category and conservation status. This will help us to understand the status of orchids in Jiangxi province. Otherwise, the question of “does mountains and water system of Ganjiang Rive and Poyang Lake affect the distribution of orchids in Jiangxi”. In my view, it is the topography and vegetation affected the diversity pattern, rather than the Ganjiang Rive and Poyang Lake. The language of this paper also needs carefully check and carefully correct by native English speaker.
There are some details need improvement.
Line 32: 1,708
Line 34: delate “people’s”
Line 34-36: this sentence need reorganized
Line 36-37: China belongs to the world. "been discovered” which tense of this?
Line 38: before “and” need an comma
Line 37-39: “new” is unnecessary in this centence
Line 39-42: “The mean reasons” and “this may lead”. They are not match in this two sentences. “The mean reasons” need match “these”
Line 44-46: this sentence need rewrite. Regulated by….., and by
Line 47, 51: before “and precipitation” need a comma
Line 53-54: what is the mean of this sentence. I think the author organized this sentence with wrong. What is the mean of “the geographical range of Jiangxi province is roughly the same as of Poyang Lake Basin”
Line 55: central China and Southern China are conflicting
Line 65: before “and” need a comma. The same as bellow
Line 127: northwestern
Line 92 and 151: change “altitude” to “elevation”. Elevation is the high from sea level. Altitude is the high get off the ground.
Round 2
Reviewer 2 Report
I was asked to evaluate the corrected manuscript. “Diversity and geographic pattern of wild orchids in nature reserves of Jiangxi, China”
Indeed, many corrections were made. However, the text did not pass English proofreading. I strongly recommend putting it in the hands of a qualified English translator!
As for the authors' responses on the Pearson correlation coefficient analyses - I stand by my point:
(the authors response) Mean while, pearson’s correlation was used to test the relationship between richness and environmental variables in many works, such as Qian, H., Deng, T., Jin, Y., Mao, L., Zhao, D., Ricklefs, R.E., 2019. Phylogenetic dispersion and diversity in regional assemblages of seed plants in China. Pro. Natl. Acad. Sci. USA 116 (46), 23192–23201
For the first paper, I cannot comment as I do not have access to it. As for the second one, I assume that the authors followed the standard procedure for correlation-regression analyses, although not every procedural step is described in the methods section.
1. Qian et al. standardized the data. There is no data standardization or normalization step in the peer-reviewed paper and the reader may doubt the validity of the test result as the Pearson coefficient is very sensitive to non-linearity and outliers and the reader was not led to believe that the authors were aware of this and took appropriate action.
(2) Qian et al. have performed Pearson correlation coefficient calculations and indeed no significance P-value is provided. Although I believe that they should have provided the P value of the test, I assume that it was significant because they performed a regression analysis.
According to all statistical books, if Pearson's r is not significant or if the scatter plot does not show a linear trend, the line should not be used for prediction and there is no justification for regression analyses. In the peer-reviewed paper, neither the P-value nor the regression analyses were provided. Therefore, I am again not convinced by the results of the correlation analyses.
Minor correction are below:
L 124 SR and SD are not diversity indexes. Authors should correct it throughout the text.
L 136 change for 85 terrestrial including 10 mycoheterotrophic species and 40 epiphytic
L 144-146 Shorten the sentence – Eight reserves (22% of those surveyed) had fewer than 10 orchid species.
L 207 Orchidaceae (capital letter and italics)
Reviewer 3 Report
Comments to the Author
The second round of review of “Diversity and geographic pattern of wild orchids in nature reserves of Jiangxi, China”, by Zhan et al.
First, I can agree the authors directly change the title to avoid the first question. It is very easy, I have not ask the author to assessment all species. The authors just need check the status on IUCN and the documents of State Key wild plant protection list. And then give the level of each species. Finally, you could present them on the map. This will improve the manuscript.
Second, I still think the paper need a native English speaker to help to improve the paper. Not only the small corrections. There are many sentences have the problem of expressing logic.
For example:
Line 13-15, Line 19-21, Line 26, and so on.
Round 3
Reviewer 3 Report
accept after check the full text again